# Risk factors and barriers to case management of neonatal pneumonia: protocol for a pan-India qualitative study of stakeholder perceptions

N Sreekumaran Nair,[1] Leslie Edward Lewis,[2] Theophilus Lakiang,[3] Myron Anthony Godinho,[3] Shruti Murthy,[3] Bhumika T Venkatesh[3]

► http://dx.doi.org/10.1136/bmjopen-2017-017616

[1]Department of Statistics and Public Health Evidence South Asia (PHESA), Manipal University, Manipal, Karnataka, India
[2]Department of Pediatrics, Kasturba Medical College, Manipal University, Manipal, Karnataka, India
[3]Public Health Evidence South Asia (PHESA), Manipal University, Manipal, Karnataka, India

**Correspondence to**
Dr N Sreekumaran Nair; sree.nair@manipal.edu

## ABSTRACT

**Introduction** India accounted for more neonatal deaths (estimated at 696 000) than any other country, as of 2015. Of these , most neonatal deaths due to infections can be attributed to pneumonia which accounts for 16% of all neonatal mortalities (2010). Despite simple, inexpensive case management strategies being available, pneumonia continues to cause significant mortality and morbidity among neonates. Understanding the perceptions and experiences of stakeholders of neonatal care can help find solutions to barriers to care and design tailored strategies for controlling neonatal pneumonia.

**Methods and analysis** A pan-India qualitative study will be conducted. Participants include healthcare providers, programme officers, academicians, representatives of non-governmental organisations/bilateral agencies and policy makers. They will be recruited purposively from rural and urban, public and private, and facility and community healthcare settings across six Indian regions. Within each region, a minimum of one state will be selected. Districts will be based on neonatal mortality indicators, and will be subject to feasibility at the time of conducting the study. We plan to conduct in-depth interviews (IDI) and focus group discussions focusing on (a) burden, (b) risk factors, (c) case management, (d) treatment guidelines, (e) barriers to case management, (f) recommendations. The number of interviews will depend on the information saturation. Interviews will be audio-recorded with prior written consent and transcribed verbatim. Principles of thematic analysis will be applied for qualitative data analysis using R package for Qualitative Data Analysis (RQDA).

**Ethics and dissemination** The protocol has been approved by the Health Ministry Screening Committee, Government of India and the Institutional Ethics Committee at the host institution. Confidentiality and privacy of the participants will be maintained. The findings of the study will be shared with all stakeholders of this research including the participants. Knowledge dissemination workshops will be conducted to ultimately transfer the evidence tailored to the stakeholders' need.

## INTRODUCTION

In recent decades in India, the neonatal mortality rate (NMR) has declined more slowly than the post-NMR, and now over

---

### Strengths and limitations of the study

► A pan-India study covering all regions of India and wide range of stakeholders of the public and private, rural and urban neonatal health system.
► Contribution of knowledge about barriers to management and control of pneumonia among neonates in the Indian context.
► Final selection of states and districts from each region will depend on the feasibility of access and willingness of participants in the district.

---

half of India's under-five deaths occur in the neonatal age group. Seventeen per cent of these are attributed to pneumonia, making it the most deadly neonatal infection.[1] In the international effort to curb child deaths, addressing neonatal pneumonia in India is clearly an issue of global concern.

Middle-income countries contribute to around 151 of the annual 156 million under-five pneumonia cases. India is the largest contributor to global pneumonia deaths, accounting for around 43 million cases per year. Neonatal pneumonia differs from childhood pneumonia in many ways, and is classified as 'early onset' or 'late onset', corresponding to the two major environments from which the infectious agent may be contracted: the intrauterine environment and birth canal; and the external environment.[2] Moreover, differences in infective agents result in a number of characteristic variants of neonatal pneumonia.[3]

The likelihood of occurrence can be augmented by risk factors and determinants. In India, economic compromise, poor immunisation status, malnutrition and indoor air pollution are some of the major risk factors for developing pneumonia.[4] Studies have previously reported that low birth weight, inadequate antenatal care,

home birthing, untrained personnel and other factors are associated with neonatal pneumonia.[5]

Effective case management is a cornerstone of pneumonia control in children. However, studies have reported barriers (eg, cost and complexities in treatment) to its implementation, especially in low-and-middle income countries (LMICs) like India, where the burden of pneumonia morbidity and mortality is high.[6] This is compounded by numerous supply issues (eg, health system, staff absenteeism, lack of infrastructure) and demand issues (eg, local socio-cultural barriers, healthcare perceptions and experiences), which plague public health systems in LMICs .[7 8]

Understanding context-specific factors and barriers for management and control of neonatal pneumonia is essential when designing effective strategies. Each stakeholder at different levels of the healthcare system (policy makers, healthcare providers, grass root level workers, informal care givers) experiences a different functional position in the healthcare delivery system. Therefore, they are able to offer in-depth, diverse and comprehensive perspectives of neonatal care, all of which are essential to attain a more complete understanding of neonatal pneumonia. The holistic understanding may help address neonatal health by involving multiple actors and multiple factors to address neonatal health, a 'complex' problem.[9] Qualitative research has successfully been used to provide evidence to inform government-stated policy priorities in middle-income countries, by promoting an understanding of local problems and solutions to develop effective and sustainable interventions.[8]

Qualitative studies have previously focused on perceptions of caregivers on barriers and factors related to pneumonia in childhood. However, empirical evidence largely exists for childhood pneumonia and neonatal sepsis. There is a need for a study which captures the variety and depth of context-specific perceptions and experiences associated with neonatal pneumonia in India's various regions.

### Aims
This qualitative study is a part of a larger mixed-methods research project, which also involves three systematic reviews on the same domains of inquiry as this study. This study will provide an in-depth and comprehensive understanding of neonatal pneumonia from stakeholders at different health system levels and settings across India. The findings of this study are planned to be synthesised with the evidence from the three reviews. The synthesised findings are anticipated to fill the evidence gap related to an understanding of the occurrence and case management of neonatal pneumonia in the Indian context. In addition, the recommendations provided by the stakeholders from their experience will serve as essential inputs to inform policy and design strategies to address neonatal health disparities more comprehensively across the diverse range of contexts present in this country.

## OBJECTIVES
1. To identify the risk factors of pneumonia in the neonatal age group (0–28 days) in the Indian context.
2. To understand pertinent barriers to the case management of neonatal pneumonia in Indian healthcare facilities.
3. To understand factors that could facilitate the control of neonatal pneumonia in India.

## METHODS AND ANALYSIS
### Research team and reflexivity
Three adequately qualified and trained researchers (female: SM; male: MG, TL) will conduct the in-depth interviews (IDI) and/or focus group discussions (FGD). SM is qualified in dentistry (BDS) and public health (MPH), MG is a medical doctor (MBBS) with public health training and TL is qualified in nursing (BSN) and public health (MPH). SM and TL had prior experience in qualitative research. All three researchers were working as research associates for this project, and received training in qualitative research from methodological experts (Faculty of Spatial Sciences, University of Groningen, The Netherlands; PRAYAS, Pune, India), especially for conducting and analysing IDI and FGD.

Regarding the language proficiencies of the interviewers, all three have native-level proficiency of English (read/write/speak) for stakeholders who would prefer English as the medium of conversation. Two of the researchers have native-level proficiency of Hindi (read/write/speak), the second official language (after English) of India, for stakeholders who would prefer Hindi as the medium of conversation. One researcher has native-level proficiency (read/write/speak) of languages spoken in f West (eg, Marathi) and South (eg, Kannada, Tamil) India; one other researcher has native-level proficiency of languages and dialects of Northeast India. Considering that this is a pan-India study, we acknowledge that there could arise a situation where language may become a barrier and introduce a bias. Thus, in specific states where we do not possess sufficient linguistic skills for an interview, for example, Telangana and Andhra Pradesh, Gujarat/Rajasthan, we have established networks and contacts who would provide us with trained interpreters on request for facilitating the interview. We will report such instances in the final manuscript after results in the section on 'Reflexivity' for information of the readers.

### Study design
A qualitative study will be conducted to explore and understand the perceptions and experiences of various stakeholders regarding the following domains of interest:
1. Risk factors associated with neonatal pneumonia in the Indian context;
2. Barriers to the case management of neonatal pneumonia;
3. Recommendations for control and management of neonatal pneumonia in India.

## Study setting

India, the largest country in South Asia, is the world's second most populous country, with about 1.2 billion people in 29 States and 7 Union Territories.[10][11] Eleven per cent of the total population is under 4 years of age.[1] Over two-thirds of the population live in rural areas, which continue to lag far behind urban India in socioeconomic progress.[12] This disparity is reflected in the neonatal health statistics: while Uttar Pradesh has an NMR of 49 deaths per thousand live births (DPTLB), which is higher than the regional aggregate for sub-Saharan Africa, Kerala boasts a single digit NMR of 7 DPTLB, resembling a considerably high-income nation.[13] Even within a state, statistics vary greatly between districts[14]; with Shrawasti and Hamirpur illustrating the contrast in Uttar Pradesh, with NMRs of 50.8 and 17.6 DPTLB, respectively.[15]

## PARTICIPANTS

Through discussions with advisory members, it was agreed that as far as possible, we would target stakeholders from every component and level of the health system. We aim to include the following stakeholders in our study:

1. Programme officers (eg, Reproductive, Maternal, Neonatal, Child and Adolescent Health (RMNCH+A)/ Integrated Management of Neonatal and Childhood Illnesses (IMNCI), state/district health officers).
2. Neonatologists, paediatricians, medical officers, neonatal nurses, general nurse midwives, auxiliary nurse midwives (ANMs) and other facility-based staff. It is anticipated that much information can be obtained on risk factors for neonatal pneumonia (hospital-acquired and ventilator-associated) and barriers to facility-based case management of neonatal pneumonia.
3. Community health workers (eg, accredited social health activists (ASHA) and state-specific equivalents), anganwadi workers. It is anticipated that community-based staff will provide valuable insights into the burden, risk factors and barriers to the case management of neonatal pneumonia, as well as other related health services and schemes available for neonatal care.
4. NGO/bilateral agencies (eg, UNICEF) representatives.
5. Academicians (eg, Indian Academy of Pediatrics members)
6. Policy makers (eg, Ministry of Health and Family Welfare representatives)

## SAMPLING STRATEGY

This study will be conducted during 2016–2017 (details on specific data collection periods have been provided in the section on 'Study status'). We will recruit participants from multiple regions within India, representing both urban and rural, as well as facility-based and community-based neonatal care. To achieve this, we adapted a classification proposed in a previous study on neonatal mortality in India, where states were grouped into six regions: North, South, Central, West, East and Northeast.[16] Within each region, a minimum of one state will be selected. Using the most recent neonatal mortality estimates available,[15] we generated a series of district-level maps of neonatal mortality indicators (published elsewhere). Within the selected state, districts were grouped by NMR from published secondary data. We identified a total of four districts in each state: two with the lowest NMR (best performers), and two with the highest NMR (worst performers) (figure 1). We plan to recruit participants from one of each of the best performing and worst performing districts, respectively. All geographical areas suspected to be conflict-prone, unsafe or inaccessible will be identified and excluded. District selection will also be subject to feasibility (eg, after contacting key informants and liaisons in identified districts and discussing with advisory committee members). From these district(s), participants from urban and rural, public and private, facility and community healthcare settings at different levels of the healthcare system will be recruited. This strategy, despite being time-intensive and labour-intensive, was developed to recruit participants from contrasting settings, allowing for a wider range of perspectives, in order to build a comprehensive picture of neonatal pneumonia in India.

## Recruitment

A purposive sampling approach will be employed to include participants from different settings and at various levels of the health system. Healthcare providers from private and public setups, at the community and facility at the primary, secondary and tertiary healthcare level will be included. Participants at different health system levels are responsible for policy making and implementation will also be identified and recruited.

To ensure successful participation, participants will be identified by liaising with governmental, non-governmental, bilateral and academic organisations, who work in maternal and child health fields in the selected regions. Contact networks of the project and advisory members will also be used to facilitate this process. The participants will then be invited through personal telephonic calls and a formal invitation letter which will include the project brief. The participants will be informed about the project goals and provided with a project brief via email. Additionally, participants will also be informed about the interviewers' background and the reasons for conducting this research.

On consenting to participate in the research, the time, location and mode (face-to-face/tele-interviews) of IDI will be ascertained through consultation with the participants. All efforts will be made to interview the participant in their natural setting. We plan to include a minimum of 24 neonatologists/paediatricians, 12 neonatal or staff nurses, 12 community health workers (ANM/ASHA), 12 medical officers and 6 policy makers. The final number of participants will depend on information saturation.

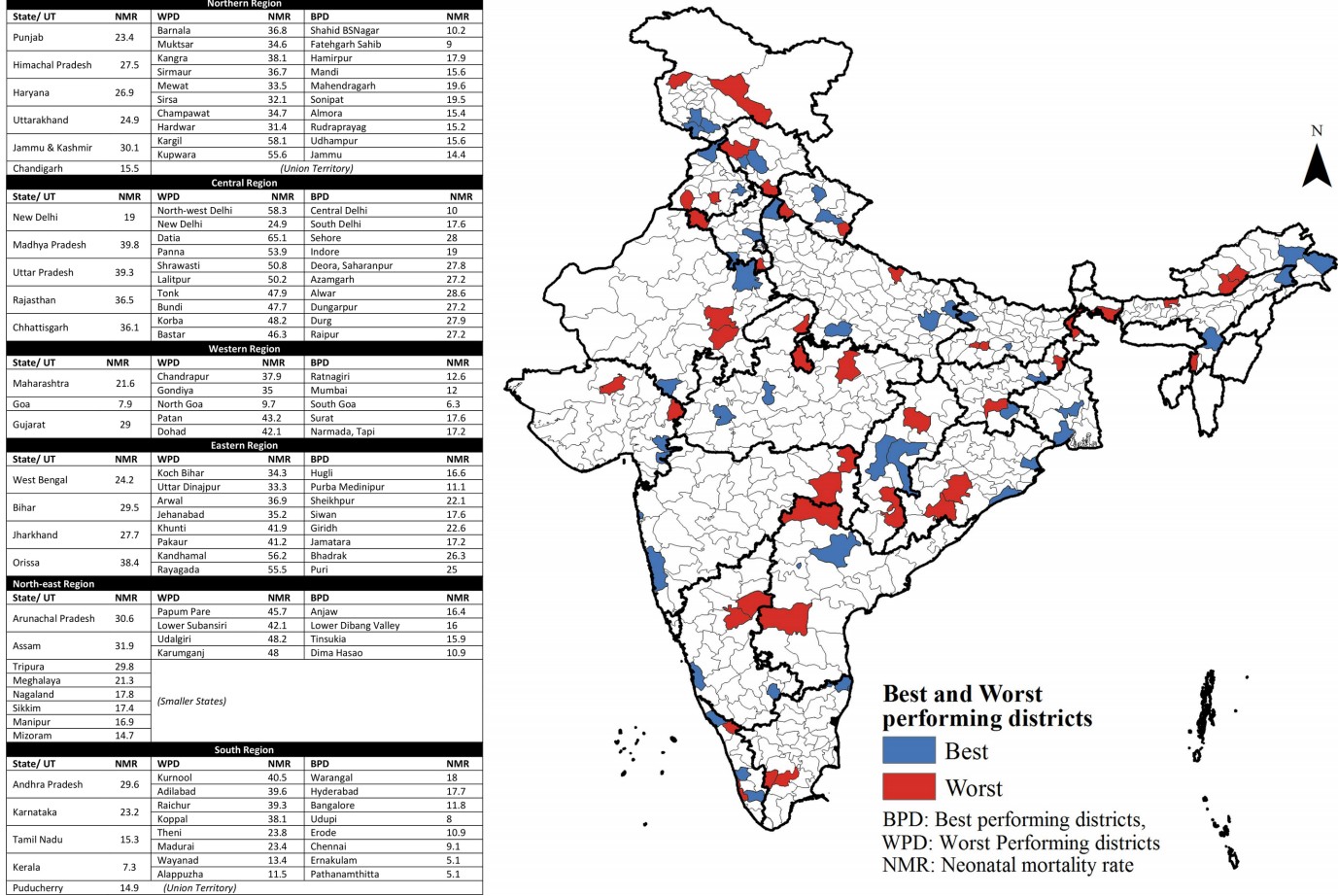

**Figure 1** A map of India depicting the two best and worst performing districts, by NMR, in each Indian state. Source: adapted, with permission, from data published by Ram et al.[15]

## Data collection methods

IDI and FGD will be used to collect data. While IDI will comprise the main mode of data collection, we anticipate that FGDs might become necessary in some instances where nurses or community health workers may be involved as participants. When a minimum of four participants of the same category (eg, nurses or ASHAs) are available and IDI may not seem feasible (eg, time constraints per participant), an FGD will be conducted. Similar guides will be used for IDI and FGD.

A detailed IDI/FGD guide will be formulated after literature review, in consultation with clinical experts and qualitative research specialists. The guide will be customised to the level of the stakeholder to allow questioning strategy to be more flexible. The IDI/FGD guide which will include the lists of issues we want to explore and will include open-ended questions on the following themes relating to neonatal pneumonia in India: (a) burden, (b) risk factors, (c) case management, (d) treatment guidelines, (e) barriers to case management, (f) recommendations. The guide for the community health workers will focus on community-acquired pneumonia in neonates and additionally include other relevant health schemes/services related to community-acquired neonatal pneumonia in India. Supporting documents will include development of (i) survey form which collects data on participant details (sociodemographic information and duration of experience), consent details and interview details and (ii) observations and field notes summary form. The guide will be pilot-tested before we proceed with data collection in order to ensure that the guide is valid and participants' understand the concept and language. Any modification required will be taken into consideration after pilot-testing of the guide, to be able to generate more in-depth information. Interviews will be conducted either through face-to-face interaction or through online sources like Skype, teleconference, etc (based on the convenience of the interviewee). During the interview, the primary researcher will lead the interview, and a second researcher will be responsible for recording the interview, making observations and taking field notes.

## Recording

Interviews will be conducted in a language which the participant is comfortable speaking. Each interview is anticipated to last for 60–90 minutes. The number of interviews per participant will depend on information saturation. For face-to-face interviews, an interview and a note taker will be present at each interview. For tele-interviews,

only the interviewer will be present at each interview. Interviews will be audio-recorded on a portable, electronic voice recorder (Sony ICD-PX440), and transferred to a computer for storage and transcription. Field notes will be maintained for every interview, which will highlight the reflections on the interview process, non-verbal cues of the interviewee as well as the initial themes that emerge as a result of the interview.

## DATA MANAGEMENT
### Data organisation
A unique identification code will be assigned to audio recording files. These codes will be based on the participant's identification code. Naming of the file will help the team to easily locate and retrieve the file whenever required for transcription. All transcripts will be named by participant ID and date and place of interview. Then the transcripts of each interview will be saved along with the informed consent of that particular participant.

### Data transcription
All interviews will be recorded (with prior consent) and transcribed verbatim (involves active and repeated listening and observation of the data collected). Interviews will first be transcribed in the language of the interview and then translated into English. The transcripts will be supplemented with field notes, if and when necessary. The English transcripts for interviews other than those in English will be back translated into that language at a later stage for quality assurance purposes. To ensure quality, 10% of the transcripts will be checked with the audio recordings by a researcher other than the interviewer.

### Data storage
A master archival log and an archival information sheet will be developed beforehand; all audio files and transcripts lists will be entered into the archival sheet. The archival sheet will contain details such as date, location, interviewer's name, name of the person who transcribed the data, etc. Backup of all transcript files to a secured external device which can be accessible only to the research team will be done.

### Data protection
Audio files, transcripts, etc will be shared only among the research team and as required by regulatory laws of the Government of India. The files will be password protected and can only be accessible by agreed members of the research team. Computer files will be kept securely and can only be shared based on the terms and conditions that the participant agreed on in the informed consent.

### Data ownership
Research team, funding agency and stakeholders will retain ownership of the data. Further clarification on data ownership will be discussed.

## Data analysis
We will follow a grounded theory approach. The transcript will be cleaned only when it has to do with the identity of the participants. Where possible, the data from the transcripts will be organised in the form of a chart or table that will allow us to easily analyse the response to each question individually, in order to make it easier to pick out concepts and themes. Thematic analysis of the transcripts will be done. To address the viewpoints and opinions collected from different stakeholders (eg, paediatricians, community health workers, policy makers), we will also analyse the responses according to the type of stakeholder.

Codes will be developed by reading and rereading the transcripts. For this, specific words/phrases/ideas/opinion/issues related to the objective of the study which occur repeatedly in the data will be identified. The interviewers will make note of the different ideas as the different responses are read through. Different codes will be explored and listed till information saturation is achieved. A code book will be developed to list and define the codes identified based on the objective of the study. Two of the research team members will code the data, and an independent qualitative research advisor of the project will check the two cycles of coding using the IDI guide as reference. The coders will then meet to discuss the coding schemes and resolve differences. Relabelling of the codes will be done if and where necessary. Next, similar codes will be grouped and categorised in a meaningful way. This will subsequently lead to generation of important categories and concepts related to the main themes of the study. Each response may be associated with one or more theme. Hence, different categories can be classified under one main overarching theme. In addition to already identified themes, themes will be derived, if any, from the categories. The data will be managed by the use of RQDA package. This software will facilitate the organisation and management of data at every stage of qualitative data analysis.

## ETHICS AND DISSEMINATION
### Ethical approval
The study will be conducted according to the principles in the Declaration of Helsinki. The protocol has been approved by the Health Ministry Screening Committee, Ministry of Health and Family Welfare, Government of India and the Institutional Ethics Committee at Manipal University, Manipal, India (ie, the host institution).

### Informed consent
Participation will be entirely voluntary. Informed consent will be mandatory for participation in the study. For face-to-face interviews, written informed consent will be obtained. For tele-interviews, verbal informed consent will be obtained. Where permission for recording is denied, handwritten notes will be taken by the researchers.[17]

## Confidentiality

Confidentiality and privacy of the participant will be maintained at every stage of the research. A unique identification code will be given to each participant. Recordings of the interviews will be assigned only the number and cannot be linked to the participant's identifiers. During data cleaning, all potential identifiers will be removed to create a 'clean' dataset.[18] The data will be kept anonymous at all stages of the study. Participants will remain unidentifiable in all transcripts, and in all reports from this study.

## Dissemination

The findings of the study will be shared with all stakeholders of this research including the participants. Knowledge dissemination workshops will be conducted with relevant stakeholders to ultimately transfer the evidence tailored to the stakeholder (eg, policy briefs, publications, information booklets, etc).

## STRENGTHS AND LIMITATIONS OF THE STUDY
### Strengths

This is a pan-India study covering all regions of India and wide range of stakeholders of the public and private, rural and urban neonatal health system. This is the first qualitative study, to the best of our knowledge, to address the issue of neonatal pneumonia in India. Thus, this study will contribute to the knowledge about prevention, management and control of pneumonia among neonates in the Indian context.

### Limitations

There are some limitations to our study. Final selection of states and districts from each region will depend on the feasibility of access and willingness of participants in the district. Second, we have not included parents/guardians as potential participants. Third, we anticipate that there may exist a perceived power difference (eg, in qualification, professional status, etc) between the interviewer and interviewee (eg, policy makers, programme managers, paediatricians), which may limit the depth of data collected. We have tried to counter this limitation with extensive training and mock interviews with similar stakeholders. Additionally, as required by consolidated criteria for reporting qualitative research (COREQ), we will report each interviewee's characteristics.

## QUALITY CONTROL AND REPORTING

Reporting of the qualitative study will be informed by two guidelines: The Standards for Reporting Qualitative Research (SRQR) [19] and the consolidated criteria for reporting qualitative research (COREQ).[20]

## STUDY STATUS

The data collection has not yet been completed for this study. The data collection has been planned in two phases to cover six regions of India. The first phase covering four regions was done from April to June 2016. Recruitment was done in the month of April–May 2016. Data collection was performed from 17 May 2016 to 5 June 2016.

The second phase is planned during July–September 2017. Recruitment and contact with stakeholders is ongoing (June 2017) for finalisation of specific dates. Data collection has been planned during third week of August 2017 and mid-September 2017 for the remaining regions.

**Acknowledgements** The authors would like to thank the following individuals for their continuous support and guidance during this process of protocol development: Dr Ajay Bailey, Assistant Professor, Faculty of Spatial Sciences, University of Groningen, The Netherlands; Dr Shrinivas Darak, Senior Researcher, PRAYAS, Pune, Maharashtra; Dr Manoj Das, Director Projects, The INCLEN Trust International, New Delhi; Dr B Shantaram Baliga, Professor, Department of Paediatrics, Kasturba Medical College, Mangalore, Karnataka; Dr K K Diwakar, Professor and Head, Department of Neonatology, Associate Dean, Malankara Orthodox Syrian Church Medical College, Kerala and Dr Unnikrishnan B, Associate Dean and Professor, Department of Community Medicine, Kasturba Medical College, Mangalore. The authors also thank Public Health Evidence South Asia (PHESA) and Manipal University, Manipal for providing the necessary institutional and infrastructural support for the project. The authors would also like to thank The INCLEN Trust International, New Delhi, and The Bill and Melinda Gates Foundation for the financial support which made this project possible.

**Contributors** SN, BV and LL conceived the research idea and reviewed the manuscript. SN and LL provided overall technical guidance. In addition, LL assisted in developing the in-depth interview (IDI) guides. TL, MG and SM designed the protocol, drafted the manuscript and developed and pilot-tested the IDI guides. All authors approved the final version of the manuscript.

**Funding** This project is supported by a grant from Bill & Melinda Gates Foundation (grant OPP1084307) to The INCLEN Trust International and sub-grant to Manipal University (subgrant INC2015GNT004).

**Disclaimer** The views expressed through this project do not necessarily represent the views of the Bill & Melinda Gates Foundation, The INCLEN Trust International, or Manipal University.

**Competing interests** None declared.

**Ethics approval** Health Ministry Screening Committee, Ministry of Health and Family Welfare (MOHFW), Government of India and Institutional Ethics Committee, Manipal University, Manipal.

**Provenance and peer review** Not commissioned; externally peer reviewed.

**Data sharing statement** Due to ethical concerns, supporting data (eg, interview transcripts) cannot be made publicly available. All other data supporting this study (eg, coding structure) will be provided as final results and as supplementary material accompanying the manuscript of the study's final results.

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
