## [Reviewer comments · BMJ Open]

ARTICLE DETAILS

TITLE (PROVISIONAL)	Risk factors and barriers to case management of neonatal pneumonia: Protocol for a pan-India qualitative study of stakeholder perceptions
AUTHORS	Sreekumaran Nair, N; Lewis, Leslie; Lakiang, Theophilus; Godinho, Myron; Murthy, Shruti; Venkatesh, Bhumika

VERSION 1 - REVIEW

REVIEWER	Charlene Rodrigues Univeristy of Oxford UK
REVIEW RETURNED	03-Jun-2017

GENERAL COMMENTS	Maybe include some epi data about burden of disease in abstract In the figure - the NMR for Goa (7.9) is lower than the BPD of 9, is this an error? Have you ensured that the you have people with the different linguistic skills to cover all regions of India, so no bias is introduced due to the interview structure of the protocol. Do community health workers also include local traditional healers/practitioners who may be seeing some babies before medical professionals, if not how do you capture their interventions?
---

REVIEWER	Kevin Baker Malaria Consortium, United Kingdom Karolinska Institute, Sweden
REVIEW RETURNED	08-Jun-2017

GENERAL COMMENTS	Overall a good protocol and a very important study. A few comments most focused on the study design and some limitations that may result from this: Page 3 Line 52 & 53: Registration of the study - it is not completely clear who the study is registered with and if this is considered a suitable registration for the study? Page 4 Line 14-22: Strengths and limitations of the study - these are listed as bullet points at the beginning of the main text or at the end of the abstract - this is not clear? These should definitely be expanded upon and are a critical part of the paper. They should address the issue with interviewing different types of stakeholders and how to analyse the different viewpoints they will get - e.g. community health workers versus academicians versus policymakers.
--

	Page 5 Line 24-25: Change text to read - Understanding the context-specific factors and barriers for management and control of neonatal pneumonia is essential when designing effective strategies. Each stakeholder..... Page 5 Line 33-34: Change text to read - This holistic understanding may help address neonatal health... Page 7 Line 8: Can you be more specific on dates? Page 9 Line 12 & 51: Sample size inclusion criteria and minimum sample requirement - these don't match and should be addressed - community health workers are missed out from the numbers listed to be interviewed while they were listed earlier as participants. Page 12: Discussion: It could be useful to include a discussion explaining more the rationale for the study design and perhaps addressing some of the limitations of the proposed study design Page 16: PRISMA checklist - check that the page numbers match as not sure the are referencing the right pages?
--	---

VERSION 1 – AUTHOR RESPONSE

Reviewer: 1

Reviewer Name: Charlene Rodrigues

Institution and Country: University of Oxford, UK

Competing Interests: None declared

Maybe include some epi data about burden of disease in abstract

Response: As recommended, two statements on epidemiological data depicting burden of disease has been now provided in abstract in the Introduction section.

In the figure - the NMR for Goa (7.9) is lower than the BPD of 9, is this an error?

Response: Thank you for the observation. This is a typographical error. The figures have been corrected as per the secondary data to South Goa: 6.3; North Goa: 9.7. The NMR remains the same of 7.9

Have you ensured that the you have people with the different linguistic skills to cover all regions of India, so no bias is introduced due to the interview structure of the protocol.

Response: Thank you for this question. We have included an additional paragraph under the section subtitled "Research team and reflexivity" to address issues regarding language proficiency of the interviewers.

Do community health workers also include local traditional healers/practitioners who may be seeing some babies before medical professionals, if not how do you capture their interventions?

Response: In our study, community health workers (CHW) do not include traditional healers/practitioners. Community health workers only include Accredited Social Health Activists (ASHA, or CHWs with similar duties and named differently in other regions), Anganwadi Worker (AWW), Skilled Birth Attendant (SBA). The scope of our study only includes health professionals (qualified) and health workers who are in the formal healthcare delivery system-either public or private. We have not planned to include traditional (unqualified) healers and practitioners. We agree that traditional healers/practitioners are important stakeholders, however, this is beyond the scope and scale of the present study.

Reviewer: 2

Reviewer Name: Kevin Baker

Institution and Country: Malaria Consortium, United Kingdom, Karolinska Institute, Sweden

Competing Interests: None declared

Overall a good protocol and a very important study. A few comments most focused on the study design and some limitations that may result from this:

Page 3 Line 52 & 53: Registration of the study - it is not completely clear who the study is registered with and if this is considered a suitable registration for the study?

Response: Thank you for this comment. The number referred to the registration of the institutional ethics committee. This specific may be confusing to the readers and hence has now been deleted.

Page 4 Line 14-22: Strengths and limitations of the study - these are listed as bullet points at the beginning of the main text or at the end of the abstract - this is not clear? These should definitely be expanded upon and are a critical part of the paper. They should address the issue with interviewing different types of stakeholders and how to analyse the different viewpoints they will get - e.g. community health workers versus academicians versus policymakers.

Response: As per the journal requirements, the section on strengths and limitations are listed at the end of the abstract. This section has been expanded upon in the section sub titled "Strengths and limitations" (Methods and Analysis) as recommended by the Editor. Additionally, a line has been added in the data analysis section to highlight how the different viewpoints will be analysed.

Page 5 Line 24-25: Change text to read - Understanding the context-specific factors and barriers for management and control of neonatal pneumonia is essential when designing effective strategies. Each stakeholder.....

Response: The recommended modification has been made on page 5, line 24-25.

Page 5 Line 33-34: Change text to read - This holistic understanding may help address neonatal health...

Response: The recommended modification has been made on page 5, line 33-34.

Page 7 Line 8: Can you be more specific on dates?

Response: We have now provided a section on "Study Status" at the end of the manuscript (in accordance to per Editor's comments) after "Quality control and reporting". On Page 7 line 8, we have provided the statement for reader interested in specific details (e.g. data collection phases and dates) to refer to the section on "study status"

Page 9 Line 12 & 51: Sample size inclusion criteria and minimum sample requirement - these don't match and should be addressed - community health workers are missed out from the numbers listed to be interviewed while they were listed earlier as participants.

Response: We have now included the numbers for community health workers in the minimum sample requirement.

Page 12: Discussion: It could be useful to include a discussion explaining more the rationale for the study design and perhaps addressing some of the limitations of the proposed study design

Response: The rationale for each different aspect of the study have been described under their respective headings in the methodology (i.e. participants, sampling strategy, recruitment, etc) where we discussed our reasoning and approach for the research design. Additionally, we have added some limitations of the proposed research design under 'sampling strategy' in methods and analysis.

Page 16: PRISMA checklist - check that the page numbers match as not sure the are referencing the right pages?

Response: The PRISMA checklist is now replaced with the COREQ checklist to display the compliance of the study protocol with COREQ guidelines.

VERSION 2 – REVIEW

REVIEWER	Kevin Baker Malaria Consortium, UK Karolinska Institute, Sweden
REVIEW RETURNED	07-Jul-2017

GENERAL COMMENTS	Thanks - this is much better and now reads very well. I have no further comments at this time.
--